# Molecular Environment Effects That Modulate the Photophysical Properties of Novel 1,3-Phosphinoamines Based on 2,1,3-Benzothiadiazole

**DOI:** 10.3390/molecules27123857

**Published:** 2022-06-16

**Authors:** Radmir M. Khisamov, Alexey A. Ryadun, Sergey N. Konchenko, Taisiya S. Sukhikh

**Affiliations:** 1Nikolaev Institute of Inorganic Chemistry, Siberian Branch of the Russian Academy of Sciences, 3 Lavrentiev Ave., 630090 Novosibirsk, Russia; khisamov@niic.nsc.ru (R.M.K.); ryadunalexey@mail.ru (A.A.R.); konch@niic.nsc.ru (S.N.K.); 2Department of Natural Sciences, National Research University—Novosibirsk State University, 630090 Novosibirsk, Russia

**Keywords:** luminescence, aggregation effects, single crystal X-ray diffraction, benzothiadiazole, phosphinoamine, TD-DFT calculations

## Abstract

We report synthesis, crystal structure, and photophysical properties of novel 1,3-phosphinoamines based on 4-amino-2,1,3-benzothiadiazole (NH_2_-*btd*): Ph_2_PCH(Ph)NH-*btd* (**1**) and Ph_2_P(E)CH(Ph)NH-*btd*, (E = O (**2α** and **2β**·thf), S (**3**), Se (**4**)). Chalcogenides **2**–**4** exhibit bright emissions with a major band at 519–536 nm and a minor band at 840 nm. According to TD-DFT calculations, the first band is attributed to fluorescence, while the second band corresponds to phosphorescence. In the solid state, room temperature quantum yield reaches 93% in the case of the sulphide. The compounds under study feature effects of the molecular environment on the luminescent properties, which manifest themselves in fluorosolvatochromism as well as in a luminescent response to changes in crystal packing and in contributions to aggregation effects. Specifically, transformation of solid **2β**·thf to solvate-free **2β** either by aging or by grinding causes crystal packing changes, and, as a result, a hypsochromic shift of the emission band. Polystyrene films doped with **2** reveal a bathochromic shift upon increasing the mass fraction from 0.2 to 3.3%, which is caused by molecular aggregation effects.

## 1. Introduction

Molecular environment effects appear as an intriguing phenomenon that give rise to unique photophysical properties of compounds both in the solid state and in solution, e.g., mechanochromism [1], solvatochromism [2], aggregation-induced emission [2], etc. These properties open up prospects for the design of smart materials for sensors, [3] optoelectronic devices [4], and biological labelling [5]. For these purposes, 2,1,3-benzothiadiazole (*btd*) derivatives have been extensively studied owing to the prospects for their applications in organic light-emitting diodes [6], luminescent sensors [7], and bioimaging [5]. Further functionalization of *btd*s provides a way to tune the electronic properties of the heterocyclic moiety that mainly contributes to the frontier orbitals. Importantly, *btd* derivatives feature secondary bonding interactions, such as chalcogen bonding and π–π stacking [8], which can govern molecular environment effects. Studies of such effects in *btd*s, e.g., aggregation-induced emission [9,10,11,12], are rapidly evolving nowadays.

On the other hand, 1,3-aminophosphines comprising {P–C–N} moiety represent a wide class of organic compounds that can be synthesized with a variation of well-known multicomponent “phospha-Mannich” reactions [13,14,15]. The introduction of various substituents is possible both at the N and P and also at the C atoms of the backbone moiety, which opens up wide possibilities for the design of such derivatives [16]. Still, a large family of 1,3-aminophosphines containing a sulphur-nitrogen aromatic substituent at the N atom remain unexplored. To date, only a couple of works have been devoted to the synthesis of related thiazole derivatives [17,18]. The corresponding 1,3-aminophosphines based on *btd* are currently unknown, although we recently reported a number of related *btd*-based phosph(III)azanes and their non-trivial photophysical properties [19,20].

In continuation of our work, we describe the synthesis and structure of a novel 1,3-phosphinoamine and its chalcogenides (O, S, and Se) bearing the *btd* substituent as a promising chromophore group as well as study their photophysical properties with special emphasis on molecular environment effects. The photophysical properties of the compounds were interpreted by means of quantum chemical methods (TD-DFT).

## 2. Materials and Methods

### 2.1. General Procedures

All manipulations for the syntheses were performed using standard Schlenk technique and a glovebox. The solvents were purified using standard technique and stored under argon atmosphere. 4-amino-2,1,3-benzothiadiazole was prepared as reported previously [21]. Benzaldehyde was distilled under argon before use. Diphenylphosphine was used as received. Elemental analyses were performed on various MICRO cube instrument for C, H, N, and S elements. IR-spectra were recorded on a Fourier IR spectrometer FT-801 (Simex) in KBr pellets (Appendix A). ^1^H NMR spectra (500.13 MHz) and ^31^P NMR spectra (202.45 MHz) were taken with a Bruker DRX-500 spectrometer in dry CDCl_3_ or C_6_D_6_ at room temperature; the solvent peak was used as the internal reference.

Thick (ca. 40 μm) films of **2** in polystyrene were prepared by coating a solution of a given amount of **2** in 100 μL of polystyrene/toluene solution (concentration of polystyrene is 36.5 mg/mL) on a glass plate, followed by drying.

The diffuse reflectance UV-vis spectra of solid samples were obtained with a Shimadzu UV-3101 spectrophotometer at room temperature. Samples for the diffuse reflectance measurements were prepared by a thorough grinding of a mixture of the compounds under study (about 0.01 mol fraction) with BaSO_4_, which was used also as a standard. Spectral dependences of the diffuse reflectance were converted into spectra of a Kubelka–Munk function. Emission and excitation spectra of solids were recorded at 298 and 77 K on a Fluorolog 3 spectrometer (Horiba Jobin Yvon, Paris, France) equipped with cooled PC177CE-010 photon detection module with a PMT R2658 photomultiplier. Excitation and emission spectra were corrected for source intensity (lamp and grating) and emission spectral response (detector and grating) by standard correction curves. For the measurements, powdered samples were placed between two nonfluorescent quartz plates. The emission decay curves were recorded on the same instrument using TCSPC (time-correlated single photon counting) technique. Laser diode was used as an excitation source. The curves were fitted by either one or two exponential decay; the fit yielded the emission lifetime values. Absolute PL quantum yields (QY) for the solids were recorded using Quanta-φ device of Fluorolog 3.

UV-vis spectra of the solutions in 1 cm quartz cuvettes in argon atmosphere and of the films in air were obtained with Agilent Cary 60 spectrometer (Agilent Technologies, Santa Clara, CA, USA). Emission and excitation spectra of the same solutions (in the transmittance mode) and films (in the reflection mode) were recorded on an Agilent Cary Eclipse spectrometer (Agilent Technologies, Santa Clara, CA, USA). The quantum yields of the samples in toluene solutions were determined relative to 4-amino-2,1,3-benzothiadiazole (QY = 12%).

For the photobleaching experiment, solid sample **3** was constantly irradiated in air conditions with a luminescent 6 W lamp for 12 h. The luminescence spectra were recorded on an Agilent Cary Eclipse spectrometer.

### 2.2. X-ray Data

Single-crystal XRD data for compounds **1**, **2β**·thf, and **4** (Appendix A) were collected with a Bruker D8 Venture diffractometer with a CMOS PHOTON III detector and IμS 3.0 microfocus source (MoK_α_ radiation (λ = 0.71073 Å), collimating Montel mirrors). Single-crystal XRD data for compounds **2α** and **3** were collected with a Bruker Apex DUO diffractometer (Karlsruhe, Germany) equipped with a 4 K CCD area detector and a graphite-monochromated sealed tube (MoK_α_ radiation). All the data were collected at 150 K. The crystal structures were solved using the SHELXT [22] and were refined using SHELXL [23] programs with OLEX2 GUI [24]. Atomic displacement parameters for non-hydrogen atoms were refined anisotropically. Hydrogen atoms were placed geometrically with the exception for those at the amino groups, which were localized from the residual electron density map and refined with the restrain on N–H bond (0.88 Å). The structures of **1**–**3** were deposited to the Cambridge Crystallographic Data Centre (CCDC) as a supplementary publication, No. 2170844-2170848.

Hirshfeld promolecular surfaces mapped over Shape Index were plotted with the Crystal Explorer (version 17.5) program [25]. The intermolecular interaction energies between pairs of molecules in the crystal were also estimated using the Crystal Explorer program with the geometry determined by single-crystal XRD with corrected C–H (1.08 Å) and N–H (1.00 Å) bond lengths at DFT B3LYP/6-31G(*d*,*p*) level.

Powder XRD data for compounds **1**, **2α**, **3**, and **4** (Appendix A) were collected at room temperature with a Bruker D8 Advance diffractometer. The data for **2β**·thf and its transformation product upon drying were collected at 150 K by a Bruker D8 Venture diffractometer (Appendix A). In all cases, CuK_α_ radiation was used.

### 2.3. Quantum-Mechanical Calculations

All calculations were performed using Orca 4.2.1 package (Wiley Interdiscip, Hudson, NJ, USA). Ground-state geometries were optimized by DFT method with PBE0-D3(BJ) functional. Geometries of S_1_ and T_1_ states were optimized with same functional by TD-DFT and unrestricted DFT methods respectively. Geometry optimizations were performed in vacuum without any constraints. Frequencies were calculated for all geometries at the same states. Optimized geometries correspond to a minimum at PES, since no negative frequencies were found. Excitation spectra were calculated by TDA-DFT with (triplet states) and without Tamm–Dancoff approximation. In all calculations def2-TZVP(-f) and auxiliary def2/J and def2-TZVP/C (for TD-DFT) basis sets were used.

### 2.4. Syntheses

#### 2.4.1. Synthesis of Ph_2_PCH(Ph)NH-btd (**1**)

4-amino-2,1,3-benzothiadiazole (400 mg, 2.65 mmol) was dissolved in tetrahydrofurane (THF; 5 mL), and benzaldehyde (0.270 mL, 2.65 mmol) and diphenylphosphine (0.462 mL, 2.65 mmol) were subsequently added while stirring. The reaction mixture was stirred overnight and volatile components were removed under vacuum. Red oil residue was treated with Et_2_O (5 mL) to give yellowish precipitate of **1**, which was filtered and washed with Et_2_O. The ether solution was concentrated under vacuum to give an extra portion of the product, which was washed with Et_2_O. Total yield 0.890 (79%). Calc. for C_25_H_20_N_3_PS (425.49): C 70.6, H 4.7, N 9.9, S 7.5. Found C 70.8, H 5.1, N 9.5, S 7.6. ^31^P{H} NMR (C_6_D_6_, *δ*, ppm): 6.05 (s). ^1^H NMR (C_6_D_6_, *δ*, ppm): 7.50 (t, 2H), 7.35 (t, 2H), 6.21 (1H, m), 6.16 (1H, d), 5.33 (1H, d). IR, cm^−1^ (assigned on the basis of DFT calculations): 3381 (N-H stretch), 3050 (C(sp^2^)-H stretch), 2876 (C(sp^3^)-H stretch), 1601–1406 (in-ring C=C stretch), 1306 (in-plane N-H and C(sp^3^)-H bending), 1122 (in-plane C(sp^2^)-H bending), 741–696 (out-of-plane N-H, C(sp^3^)-H and C(sp^2^)-H bending).

Crystals of compound **1** suitable for single-crystal XRD study were grown by recrystallization of the product from Et_2_O.

#### 2.4.2. Synthesis of Ph_2_P(O)CH(Ph)NH-btd (**2**)

4-amino-2,1,3-benzothiadiazole (400 mg, 2.65 mmol) was dissolved in THF (5 mL), and benzaldehyde (0.270 mL, 2.65 mmol) and diphenylphosphine (0.462 mL, 2.65 mmol) were subsequently added while stirring. The reaction mixture was stirred overnight and volatile components were removed under vacuum. The residue was dissolved in acetontrile and excess of H_2_O_2_ (30%) was added. The reaction mixture was stirred overnight to give fine yellow precipitate, which was filtered and washed with acetonitrile and Et_2_O.

Yield 0.817 g (70%). Calc. for C_25_H_20_N_3_OPS (441.48): C 68.0, H 4.6, N 9.5, S 7.2. Found C 68.0, H 4.7, N 9.6, S 8.4. ^31^P{H} NMR (CDCl_3_, δ, ppm): 31.60 (s). ^1^H NMR (CDCl_3_, δ, ppm): 7.88–7.84 (2H, m), 7.56–7.42 (6H, m), 7.30 (2H, td), 7.26–7.23 (1H + solvent), 7.21–7.19 (2H, m), 7.16–7.12 (3H, m), 6.47 (1H, t), 6.28 (1H, d), 5.39 (1H, dd). IR, cm^−1^ (assigned on the basis of DFT calculations): 3393 (N-H stretch), 3059 (C(sp^2^)-H stretch), 2870 (C(sp^3^)-H stretch), 1605–1404 (in-ring C=C stretch), 1304 (in-plane N-H and C(sp^3^)-H bending), 1184 (P=O stretch), 1115 (in-plane C(sp^2^)-H bending), 736–696 (out-of-plane N-H, C(sp^3^)-H and C(sp^2^)-H bending).

Crystals of compound **2α** were grown by recrystallization of the product from ethanol.

Crystals of **2β**·thf were grown by diffusion of *n*-hexane to saturated THF solution of **2**.

#### 2.4.3. Synthesis of Ph_2_P(E)CH(Ph)NH-btd (E = S (**3**), Se (**4**))

4-amino-2,1,3-benzothiadiazole (200 mg, 1.32 mmol) was dissolved in THF (5 mL), and benzaldehyde (0.135 mL, 1.32 mmol) and diphenylphosphine (0.230 mL, 1.32 mmol) were subsequently added while stirring. After stirring of reaction mixture overnight, solid S_8_ (for **3**; 42 mg, 0.16 mmol) or red Se (for **4**; 104 mg, 1.32 mmol) were added. The reaction mixtures were refluxed for 2 h and cooled to room temperature. Concentration under vacuum gave crystalline precipitates, which were filtered and washed with THF. The solutions were concentrated under vacuum to give an extra portion of the products, which were washed with THF.

#### 2.4.4. Ph_2_P(S)CH(Ph)NH-btd (**3**)

Yield 0.569 g (94%). Calc. for C_25_H_20_N_3_PS_2_ (457.55): C 65.6, H 4.4, N 9.2, S 14.0. Found C 65.7, H 4.7, N 9.0, S 13.9. ^31^P{H} NMR (CDCl_3_, δ, ppm): 51.72 (s). ^1^H NMR (CDCl_3_, δ, ppm): 7.97–7.92 (2H, m), 7.50–7.37 (6H, m), 7.27–7.21 (4H, m), 7.12–7.10 (3H, m), 7.04 (2H, t), 6.93 (1H, t), 6.28 (1H, d), 5.62 (1H, dd). IR, cm^−1^ (assigned on the basis of DFT calculations): 3366 (N-H stretch), 3058–3027 (C(sp^2^)-H stretch), 2861 (C(sp^3^)-H stretch), 1966–1701 (overtones), 1605–1410 (in-ring C=C stretch), 1307 (in-plane N-H and C(sp^3^)-H bending), 1101 (in-plane C(sp^2^)-H bending), 748–655 (out-of-plane N-H, C(sp^3^)-H and C(sp^2^)-H bending).

#### 2.4.5. Ph_2_P(Se)CH(Ph)NH-btd (**4**)

Yield 0.538 g (81%). Calc. for C_18_H_14_N_3_OPS (351.36): C 61.5, H 4.0, N 12.0, S 9.1. Found C 61.8, H 3.9, N 11.8, S 9.3. ^31^P{H} NMR (CDCl_3_, *δ*, ppm): 48.19 (s). ^1^H NMR (CDCl_3_, *δ,* ppm): 7.92–7.98 (4H, dd), 7.49–7.37 (6H, m), 7.28–7.22 (1H + solvent, m), 7.13–7.10 (3H, m), 7.04–6.99 (3H, m), 6.30 (1H, d), 5.66 (1H, dd), 3.73 (1H, t). IR, cm^−1^ (assigned on the basis of DFT calculations): 3347 (N-H stretch), 3056–3027 (C(sp^2^)-H stretch), 2859 (C(sp^3^)-H stretch), 1601–1408 (in-ring C=C stretch), 1307 (in-plane N-H and C(sp^3^)-H bending), 1099 (in-plane C(sp^2^)-H bending), 748–644 (out-of-plane N-H, C(sp^3^)-H and C(sp^2^)-H bending).

Crystals of compounds **3** and **4** were grown by recrystallization of the products from hot THF.

## 3. Results and Discussion

### 3.1. Syntheses and Crystal Structures

One-pot condensation of diphenylphospine, benzaldehyde, and 4-amino-2,1,3-benzothaidiazole in anhydrous THF at ambient conditions yielded α-aminophosphine **1** (Figure 1). The corresponding chalcogenides **2**–**4** were obtained by in situ oxidizing of **1** with H_2_O_2_ or elemental chalcogens. For the reaction with H_2_O_2_, the solvent was removed in vacuum and replaced by acetonitrile due to risk of formation of explosive peroxides. Crystalline compounds were purified by recrystallization from Et_2_O or THF and were characterized by ^1^H, ^31^P NMR spectroscopy and elemental analysis.

In the special case of the phosphinoxide derivative, we isolated two crystalline modifications, **2α** and **2β**·thf. The first one is formed as plate-like crystals upon concentration of THF, ethanol, or aceronitrile solution, or upon addition of *n*-hexane to THF solution (Appendix A). The second modification, **2β**·thf, is formed as needle crystals upon diffusion of *n*-hexane to saturated THF solution in a narrow period of time. Aging of the crystals of **2β**·thf under the mother liquor for several days causes transformation to compound **2α**. Fine crystalline phase **2β**·thf is stable on air for ca. 15 min; after that, it starts losing solvate thf molecules, gradually transforming through a number of related phases to a structurally different solvate-free phase, **2β**, as evidenced by powder XRD analysis (Figure 1). The complete transformation takes place within a week. Unfortunately, the poor quality of the **2β** crystals did not allowed us to solve the structure, although Pawley refinement using powder XRD gave crystal lattice with expectedly smaller unit cell volume compared to the pristine **2β**·thf (Appendix A). Note that the XRD pattern of aged **2β** does not correspond to that of **2α**. The crystal-to-crystal transformation between **2α** and **2β** does not proceed, since the mutual orientation of the molecules in **2α** and **2β**·thf strongly differs (see the discussion below).

According to the XRD analysis, compound **2α** crystallizes in the chiral space group *P*2_1_; the molecule possesses chirality at the central C atom determined as *R* for the selected single crystal. Since the starting reagents are achiral, both *S* and *R* enantiomers likely form equally, so that the particular chirality of the crystals **2α** is random. Compound **2β**·thf crystallizes in the achiral space group *Pbca*, comprising both *S* and *R* molecules in the structure. Molecular geometries of the phosphinoamines in **2α** and **2β**·thf are quite similar (Figure 2), while their mutual orientation in the crystals is different. In **2α**, the molecule lies above translationally equivalent one thus arranging in –A–A– sequence along *a* direction (Figure 3a). In **2β**·thf, the corresponding molecules relating via the glide plane arrange in –A–B– stack. According to Hirshfeld surface analysis [25], both molecules exhibit similar shape; the area of contacting surfaces for the two molecules are similar for **2α** (ca. 70 Å^2^) and **2β**·thf (ca. 80 Å^2^), while in the latter, they interact more strongly with each other (Appendix A). The Hirshfeld surfaces feature a convex induced by the O atom (Figure 3b); {CH} moieties of the aryl fragments organize around the convex, and {CH} of the methylene group points straight at the O atom.

Sulphide and selenide compounds **3** and **4** are isostructural with each other crystallizing in *P*2_1_/*n* space group. Molecular geometries of **2**–**4** are similar (Appendix A): the torsion angle O–P–C–N that determines the conformation varies in a narrow range of 55.9 (for **2β**·thf)–62.8° (for **3**). However, the crystal packing of **3** and **4** differs from that of **2α** and **2β**·thf: the central methylene group of the neighbouring molecule does not point at the chalcogen atoms (Figure 3a,b) due to much larger Van der Waals radius of S and Se atoms (1.80 and 1.90 Å) comparing to O (1.52 Å). In other words, the neighbouring molecules in **3** and **4** do not arrange one above the other for steric reasons.

### 3.2. Photophysical Properties

The UV-Vis spectra were measured of **1**–**4** in the solid state and in toluene solution (Figure 4a,b, Table 1). All the compounds exhibit a similar shape of the spectra that reveal two broad bands. According to TD-DFT calculations for the single molecules (Appendix A), the long wavelength band corresponds to *S*_0_–*S*_1_ transition between HOMO and LUMO orbitals (Figure 5 and Appendix A). The transition has locally excited character with a notable contribution of intramolecular charge transfer from non-bonding orbitals of P (for **1**) or chalcogens (for **2**–**4**). Upon going from **2** to **4**, localization of HOMO orbitals on chalcogenide atom increases, i.e., the non-bonding character of HOMO-LUMO transition increases by the rise of chalcogen weight. The maximum of the bands in the solid state (445–460 nm) bathochromically shifts compared to the solution (403–409), which implies influence of the immediate environment of the molecules on the photophysical properties of **1**–**4**. Notably, the calculated wavelength for *S*_0_–*S*_1_ transition well agrees with the experiment in the case of the solutions of **2**–**4** and is slightly overestimated for the solution of **1**. The short wavelength band at ca. 310 nm is attributed to π–π* transition in the *btd* moiety.

Compounds **1**–**4** at room temperature reveal photoluminescence (PL) both in the solid state and toluene solution, characterized by a major band with the excitation-independent maximum of 519–538 nm (Figure 6a,b). According to TD-DFT calculations, the band corresponds to *S*_1_–*S*_0_ transition between frontier orbitals; the calculated wavelengths well agree with the experiment (Table 1 and Appendix A). The frontier orbitals in *S*_1_-geometry state resemble those in *S*_0_ state, featuring for the HOMO a slight loss of the electron density at the chalcogen atom in favour of the phenyl moiety at the C atom (Appendix A). The nature of the chalcogen at P atom slightly affects the position of the band. The PL intensity in toluene rises in the series **1**–**2**–**3** and then falls for **4**. This tendency persists for absolute quantum yields for the solid compounds (Table 1). The reduced intensity for **4** is likely associated with the heavy Se atom effect that causes emission quenching. The emission kinetics for the major band of the solids (Appendix A) can be fitted either one (in the case of **4**) or two exponential decay (in the case of **2** and **3**; for **1**, we failed to evaluate the parameters due to the poor emission intensity). The presence of two components may indicate the existence of different emission aggregates that originate from some features of solids, e.g., defects and surface characteristics, inclusions, among others. In all the cases, the emission lifetime lies in a nanosecond range of 4–20 ns implying the major band corresponds to a fluorescence. We have performed the photobleaching experiment on the example of solid **3**: upon continuous irradiation of the compound with a visible light on air, the emission intensity gradually drops by ca. 10% in 12 h (Appendix A).

In the solutions, the excitation spectra resemble the absorption spectra, indicating a straightforward relaxation process to the lowest excited state. In the solid state, the relaxation process is less straightforward due to the appearance of solid-state effects, e.g., a relatively strong electron–phonon coupling [26], and extrinsic features (inclusions and defects). As a result, nonradiative decay rates for the relaxation from different excited states can vary, which leads to some differences in the absorption and excitation spectra, i.e., the flattened profile of the latter.

Apart the major band, the PL spectra of solid **2**–**4** reveal a weak but notable band in the IR range peaked at ca. 840 nm. Although the experimental emission lifetime was not determined for the IR band due to the poor emission intensity, we can attribute the minor band to a phosphorescence. Indeed, the calculated wavelength for *T*_1_–*S*_0_ transition well agrees with the experiment (Table 1 and Appendix A). The intersystem crossing (ISC) promotion to *T*_1_ state is likely achieved from *S*_1_ state, which is facilitated by the presence of n-electron chalcogenide groups at the P atom. However, due to the similar nature of *S*_1_ and *T*_1_ states (Appendix A), in accordance with the El’Sayed rule [27], the ISC is rather inefficient, which results in a low intensity of the emission at 840 nm. Upon cooling from room temperature to 77K, the emission bands for the solid compounds do not change their position while showing some increase in intensity (Appendix A). Note that in the toluene solutions, no signal above the noise in the IR range was observed. The quenching of the phosphorescence upon going from the solid state to the solution can be a consequence of a facilitated intramolecular motion in the latter, which results in increased probability of radiationless transitions. Thus, we interpret the phenomenon of induced solid-state phosphorescence in terms of Restriction of Intramolecular Motion (RIM), which is proposed as a mechanism of aggregation-induced emission (AIE) [28].

Interestingly, pristine crystalline compound **2β**·thf shows a bathochromic shift of the major band by 10 nm compared to **2α**. Upon drying, **2β**·thf transforms to solvate-free compound **2β**, which reveals the same position of the band as **2α** (Figure 6a and Appendix A). Such a behaviour can be rationalized by the loss of contacts of the phosphinoamine with thf molecules and/or by changing its contacts with the neighbouring phosphinoamine molecules, which cause slight changes in the electronic structure of the molecule.

To further study the influence of the immediate molecular environment on the photophysical properties, we measured UV-Vis and PL spectra of solutions of **1** in *n*-hexane, THF and toluene (Figure 7a,b). PL maxima show increase in the maxima in the series: n-hexane (520 nm), toluene (537 nm,) and THF (553 nm). This tendency agrees with the rise in polarity of the solvents and corresponds to the positive solvatochromism [29]. In addition, the THF: *n*-hexane (*v*/*v* = 1:3) mixture for **1** gives the intermediate maximum of 536 nm between those in pure *n*-hexane and THF (Appendix A).

In the case that the luminescent properties are sensitive to the immediate molecular environment, a substance can exhibit aggregation effects upon increasing its doping concentration in a matrix. To find whether these effects appear in our case, we prepared thick polymeric polystyrene films doped with **2** with the mass fraction varied from 0.2 to 3.3%. As expected, the films show a linear dependence of the absorption at 425 nm on the fraction of **2** (Figure 8a). The luminescence spectrum of the film with the highest doping resembles that of the solid **2α** while showing a gradual shift of the band from 535 to 516 nm (Figure 8b) with the doping decrease. This phenomenon is apparently associated with the effect of aggregation of phosphinoamine molecules in films, which results in a bathochromic shift with increasing concentration. Such a shift can accompany aggregation effects as reported in works on the aggregation-induced phenomenon [30]. Positions of the emission band do not change after a week. We can exclude that the shift of the band is associated with the concentration quenching effects, since the excitation at the short wavelength side of the emission band (e.g., at 480 nm) still contributes to an intense emission, being even higher for high-doped films than for low-doped ones (Appendix A).

## 4. Conclusions

In summary, we synthesized and characterized a novel 1,3-phosphinoamine (**1**) bearing 4-amino-2,1,3-benzothaidiazole moiety and its oxide (**2**), sulphide (**3**), and selenide (**4**). For **2**, two pseudopolymorphs, **2α** and **2β**·thf, were prepared; the latter gradually loses solvate thf molecules upon aging in air, transforming to solvate-free **2β**. Compounds **1**–**4** reveal a major band at 519–538 nm both in the solid state and in toluene solution. Solid samples **2** and **3** feature bright emission with room-temperature absolute quantum yield of 93% for the latter. In toluene solutions, the quantum yield for **2** and **3** is close to 100%. TD-DFT investigation suggests that the band is attributed to *S*_1_–*S*_0_ relaxation of a fluorescent nature; the corresponding HOMO–LUMO transition tends to increase charge-transfer contribution with the rise of chalcogen weight. This is a probable reason why solid compounds **2**–**4**, apart from the major band in the visible region, reveal a weak band in the IR range attributed to *T*_1_–*S*_0_ phosphorescence.

We studied the effects of the immediate molecular environment on the following luminescence properties: (1) Molecules of **1** show luminous sensitivity to the nature of solvent molecules (hexane, toluene, and THF), exhibiting positive fluorosolvatochromism in the corresponding solutions. (2) Phase transformation of solid **2β**·thf to **2β** either by aging or by grinding causes crystal packing changes, and, as a result, shift of the emission band; this opens up perspectives for stimuli-responsive effects. (3) Polystyrene films doped with 0.2 to 3.3% of **2** differ in the luminescence colour caused by molecular aggregation effects. This motivates further design of luminescent benzothiadiazoles that pave the way for developing novel fluorochromic materials.

## Data Availability

Not applicable.

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
