# Peer review of "Molecular Environment Effects That Modulate the Photophysical Properties of Novel 1,3-Phosphinoamines Based on 2,1,3-Benzothiadiazole"

_molecules, 2022, doi:10.3390/molecules27123857_

Round 1

Reviewer 1 Report

In this study, the authors report the synthesis, structure, luminescence, and photophysical parameters of a series of new organic compounds based on 4-amino-2,1,3-benzothiadiazoles. The authors carried out a nice structural characterization by crystal X-ray diffraction and DFT. Moreover, they reported the influence of the molecule processing (powder or in several solvents) on the photoluminescent properties and gave reasonable explanation to all observed changes. Although all the luminescent phenomena observed regarding the changes of the molecular environment are well-known for organic molecules, the authors took advantage of this knowledge to explain the phenomena in this new series of molecules. Due to those reasons I recommend publication of this study after minor revisions that I highlight in the sequence:

1) Authors should provide all the assignments in FTIR presented in Figure S1;

2) I could not find the emission spectra measured at 77 K, could the authors provide them in the supplementary material?

3) Could the authors provide how they measured the emission lifetime and also show the emission decay curves? Why were two lifetime values found? Probably there is no impurity in the systems, so could it be due to different aggregation states in the powder system?

4) It would be interesting from the application point of view if the authors show photobleaching studies at least for the best system (larger quantum yield). Authors could do that by measuring the emission intensity upon continuous light exposition for a certain time (12 hours for instance).

Author Response

We thank the Reviewer for careful reading of the manuscript and for helpful comments. We have revised the manuscript as follows:

1) Authors should provide all the assignments in FTIR presented in Figure S1;

We have included the assignment of the bands in the experimental section.

2) I could not find the emission spectra measured at 77 K, could the authors provide them in the supplementary material?

We have included the corresponding spectra in the new version of supplementary materials (Fig. S18)

3) Could the authors provide how they measured the emission lifetime and also show the emission decay curves? Why were two lifetime values found? Probably there is no impurity in the systems, so could it be due to different aggregation states in the powder system?

We have included the information about the measurement in the experimental section: “The emission decay curves were recorded on the same instrument using TCSPC (Time-Correlated Single Photon Counting) technique. Laser diodes were used as an excitation source. The curves were fitted by either one or two exponential decay; the fit yielded the emission lifetime values.” We also have included the corresponding curves in the new version of supplementary materials (Fig. S16). The point about two lifetime values as a consequence of the presence of different aggregation states is indeed interesting. We have included this hypothesis in the text: “The presence of two components can indicate the existence of different emission aggregates that originate from some features of solids, e.g. defects, surface characteristics, inclusions, etc.”

4) It would be interesting from the application point of view if the authors show photobleaching studies at least for the best system (larger quantum yield). Authors could do that by measuring the emission intensity upon continuous light exposition for a certain time (12 hours for instance).

This is indeed an interesting point. We have carried out a preliminary photobleaching study of compound 3 featuring the highest quantum yield. The emission intensity gradually drops by ca. 10% under continuous irradiation in air in 12 hours. We have included this information in the manuscript and included the spectral data in the supplementary materials (Fig. S17).

Reviewer 2 Report

The authors presented for publication and article in which they study the various modulations of photophysical properties in 4 different compounds. The research is novel, interesting and properly conducted. There are few issues I believe should be addressed and I recommend the manuscript to be published after minor revision.

1.       The details of quantum yield measurements are missing from the Materials and Methods

2.       No reference to Scheme 1 in the text

3.       Fig. 2 does not illustrate XRD analysis, as suggested in the text, but rather the solved crystal structures

7.       References missing to described phenomena e.g. bathocromic shift, solvatochromism, El’Sayed rule.

I I would also suggest authors some questions to be adressed in the manuscript:

4.       What is the origin of short wavelength band in the absorption spectrum?

5.       Why do the excitation spectra differ from the absorption spectra?

6.       Since 2β has the same PL characteristic as 2α, could it be that 2β-thf over time transitions to 2α?

Author Response

We thank the Reviewer for careful reading of the manuscript and for helpful comments. We have revised the manuscript as follows:

  1. The details of quantum yield measurements are missing from the Materials and Methods

We have included the corresponding information in the manuscript: “Absolute PL quantum yields (QY) for the solids were recorded using Quanta-φ device of Fluorolog 3. The quantum yields of the samples in toluene solutions were determined relative to 4-amino-2,1,3-benzothiadiazole (QY = 12%).”

  1. No reference to Scheme 1 in the text

We have included the reference to Scheme 1.

  1. Fig. 2 does not illustrate XRD analysis, as suggested in the text, but rather the solved crystal structures

We have moved the reference to Fig. 2 to the appropriate phrase.

  1. References missing to described phenomena e.g. bathocromic shift, solvatochromism, El’Sayed rule.

We have added the corresponding references in the new version of the manuscript: El'Sayed rule [27]; positive solvatochromism [29]; bathochromic shift [30].

  1. What is the origin of short wavelength band in the absorption spectrum?

The short wavelength band at ca. 310 nm attributes to π–π* transition in the btd moiety. We have included this information in the manuscript.

  1. Why do the excitation spectra differ from the absorption spectra?

In the solutions, the excitation spectra resemble the absorption spectra, indicating a straightforward relaxation process to the lowest excited state. In the solid state, the relaxation process is less straightforward due to the appearance of solid-state effects, e.g. a relatively strong electron-phonon coupling [26], and extrinsic features (inclusions and defects). As a result, nonradiative decay rates for the relaxation from different excited states can vary, which leads to some differences in the absorption and excitation spectra, viz. flattened profile of the latter. We have included this discussion in the manuscript.

  1. Since 2β has the same PL characteristic as 2α, could it be that 2β-thf over time transitions to 2α?

2β-thf does not transform to 2α upon drying, as evidenced by powder XRD data: the XRD pattern of aged 2β does not correspond to that of 2α. The crystal-to-crystal transformation between 2α and 2β proceeds unlikely, since the mutual orientation of the molecules in 2α and 2β-thf strongly differs (Fig. 3). We have included this information in the text.